# Autologous Matrix-Induced Chondrogenesis (AMIC) for Focal Chondral Lesions of the Knee: A 2-Year Follow-Up of Clinical, Proprioceptive, and Isokinetic Evaluation

**DOI:** 10.3390/jfb13040277

**Published:** 2022-12-06

**Authors:** Paweł Bąkowski, Kamilla Grzywacz, Agnieszka Prusińska, Kinga Ciemniewska-Gorzela, Justus Gille, Tomasz Piontek

**Affiliations:** 1Department of Orthopedic Surgery, Rehasport Clinic, 60-201 Poznań, Poland; 2Institute of Bioorganic Chemistry Polish Academy of Sciences, 61-704 Poznań, Poland; 3Department of Orthopaedic and Trauma Surgery, University Hospital Schleswig-Holstein, Campus Luebeck, 23562 Luebeck, Germany; 4Department of Spine Disorders and Pediatric Orthopedics, University of Medical Sciences, 60-201 Poznań, Poland

**Keywords:** AMIC, focal chondral lesions, cartilage chondrogenesis, collagen scaffolds, biomaterials, tissue engineering

## Abstract

(1) Background: The autologous matrix-induced chondrogenesis (AMIC) is a bio-orthopedic treatment for articular cartilage damage. It combines microfracture surgery with the application of a collagen membrane. The aim of the present study was to report a medium-term follow-up of patients treated with AMIC for focal chondral lesions. (2) Methods: Fourty-eight patients treated surgically and 21 control participants were enrolled in the study. To evaluate the functional outcomes, the proprioceptive (postural stability, postural priority) and isokinetic (peak value of maximum knee extensor and flexor torque in relation to body mass and the total work) measurements were performed. To evaluate the clinical outcomes, the Lysholm score and the IKDC score were imposed. (3) Results: Compared to the preoperative values, there was significant improvement in the first 2 years after intervention in the functional as well as subjective outcome measures. (4) Conclusions: AMIC showed durable results in aligned knees.

## 1. Introduction

The main function of cartilage is to cover the joint surfaces of the long bones to prevent their wear. Elastic tissue is composed of the chondrocytes which produce prominent amounts of extracellular matrix, rich in proteoglycan, glycosaminoglycans, proteoglycans, and collagen. The cartilage is avascular and aneural, which results in a limited self-healing capability, and therefore the chondral defects are debilitating, often requiring surgical management [1]. For that reason, one of the greatest challenges nowadays for orthopaedic surgeons around the world is the treatment of cartilage injuries. Many recognized surgical procedures have been developed in recent decades, e.g., microfractures, joint surface debridement (which could be further supported with microfracturing) [2], and osteochondral autograft [3] or allograft transplants [4]. It has been observed that the outcomes of the microfracturing cartilage repair have a tendency to regress with time; on the other hand, satisfactory long-term cartilage functionality was observed in patients who underwent the other procedures [5]. Notwithstanding, all of the techniques mentioned could lead to undesirable effects, e.g., donor-site morbidity or auto- and allograft mismatches. Hence, there is growing enthusiasm for cell-based orthobiological techniques to repair chondral defects, which would lead to hyaline cartilage repair.

The autologous matrix-induced chondrogenesis (AMIC) is an innovative, fully arthroscopic orthobiological method of the articular cartilage reconstruction that harnesses the body’s natural potential to rebuild the focal cartilage and osteochondral lesions with an area exceeding 1–2 cm^2^ [6]. AMIC is combined with microfractures, covering the lesions with a resorbable two-layer collagen membrane to sustain the subsequent blood supply. This creates an appropriate environment for the bone marrow-derived mesenchymal stem cells arising from the subchondral bone to differentiate into chondrocytes.

Functional evaluation is extremely important in making therapeutic decisions. Despite the long-lasting research, there is no single, universal scale that would assess the functional changes in the knee joint after cartilage reconstruction surgery. The proprioception or neuromuscular control is an important part of the knee joint function [7]. It provides the nerve stimulation from the joints, muscles, tendons, and deep tissues, which are processed in the central nervous system and provide the information about the position of the joint, movement, vibration, and pressure. The muscles’ strength and endurance are the second most important elements in the function of the knee joint [8]. Rapid muscular atrophy occurs immediately after the surgery as a response to pain, inflammation, and immobilization. Even after a short immobilization, a decrease in muscle strength occurs; in addition, the muscles show greater fatigue, which is translated as a decrease in muscle endurance. Clinical assessments provide a large amount of measurable information about the joint, such as contour disturbances, limb axis deviations, changes in the color and temperature of the skin, as well as joint mobility and stability. This study does not take into account the patient’s own feelings, such as joint pain, a sense of fitness or functional limitations, which is why the importance of the patient’s subjective assessment is increasingly emphasized [9]. Without such an evaluation and its thorough analysis, it is impossible to obtain a complete picture of the healing of the surgically treated joint.

There are several studies that have evaluated the clinical outcomes of the AMIC for the treatment of focal chondral defects of the knee [10,11,12,13]. However, none of them tackled either proprioception or muscle strength and endurance assessments. We have therefore noticed a need to evaluate not only subjective outcomes, but also objective ones. We have compared the results to those for untreated participants.

## 2. Materials and Methods

### 2.1. Recruitment Process

All patients treated with an AMIC procedure at the Rehasport Clinic, Poznań (Poland), for full-thickness cartilage tears of the knee between 2011 and 2013 were retrospectively reviewed. All patient-derived data have been fully anonymized and collected according to the clinic’s recommendations. The following inclusion criteria were used: (1) magnetic resonance imaging (MRI) evidence; (2) the arthroscopic reconstruction of the knee joint cartilage using the AMIC method; (3) implementation of the RSC rehabilitation protocol; and (4) informed consent to participate in the study. The exclusion criteria included postoperative complications, problems during rehabilitation, and the inability to perform the assessment on any of the proposed dates.

A total of 61 patients were initially selected for the study. Of them, 13 were not eligible to attend the whole assessment. A total of 48 patients were operated on with AMIC, and 26 patients were enrolled in an assessment 24 months after the surgery. Of the 26 patients, 34.6% (*n* = 9) were women and 65.4% (n = 17) were men. In 18 patients (69%), the lesion was located on the dominant limb. The mean age was 44 y.o. (±11) and the mean BMI was 26 kg/m^2^ (±4).

A control group of healthy volunteers consisted of 21 participants. The exclusion criteria for the control group were: (1) previous surgery of the knee joint; (2) pain in the knee joint; (3) health contraindications for the biomechanical assessment; and (4) lack of informed consent to participate in the study. The comparison of the research group with the control group is presented in Table 1.

### 2.2. Surgical Technique

All the surgeries were performed in the same fashion by one experienced surgeon (T.P.) according to the previous report, which describes the AMIC procedure in details and is justified with intraoperative images of the procedure [14]. Summarily, a qualifying diagnostic arthroscopy was performed with classic anteromedial and anterolateral access. Then, the damaged tissue was removed to obtain stable cartilage borders, and the size of each defect was estimated. The chondral defect was cleaned of fibrous tissue and osteophytes with one punch (in cases of smaller cartilage defects, up to 11 mm in diameter) or more (defects larger than 11 mm in diameter). The second part of the procedure was executed under dry arthroscopic conditions. Numerous bores were drilled at 5-mm intervals in the subchondral layer, and Chondro-Gide membrane circles (Figure 1) were placed in the reconstruction area with the porous surface of the membrane facing the bone surface, using a fibrin glue. The stability of the membrane was checked by repeatedly flexing and extending the knee under direct vision. Next, the arthroscopic access wounds were closed.

### 2.3. Rehabilitation Protocol

Patients were provided with an adjustable-angle orthosis that stabilized the knee at a 15° for the first 24 h after surgery.

Important elements of the rehabilitation protocol of patients after the AMIC cartilage reconstruction were introduced gradually: exercises to improve muscle performance; flexibility exercises—range of motion in the joint and muscle flexibility; exercises for neuromuscular control (proprioception); functional exercises; exercises to correct biomechanical abnormalities; exercises to maintain the efficiency of the cardiovascular system; and psychotherapy. A detailed description of the rehabilitation program is presented in Table 2.

### 2.4. Evaluations

The functional and the clinical outcomes were collected. All patients were assessed postoperatively.

The visual-proprioceptive control was evaluated by measuring the postural strategy (PS) and the postural priority (PP) parameters with the Delos Postural Proprioceptive System. We have implemented and tested dynamic and static Riva tests, as described in [15]. The results for the PS parameter were classified as follows (according to Riva standards [16]): excellent (0.0–1.0°), very good (1.0–2.5°), good (2.5–5.0°), sufficient (5.0–9.0°), or insufficient (>9.0°). The results for the PP parameter were classified as correct (60%), incorrect (40–60%), or bad (<40%).

The isokinetic evaluation of the extensor and flexor muscles was performed using a Biodex 3 dynamometer, where the peak value of maximum knee extensor and flexor torque at 60°/s velocity (Peak Torque, PT), the peak value of maximum knee extensor and flexor torque in relation to body mass at 60°/s velocity (Peak Torque/Body Weight, PT/BW), and the total work of the knee extensors and flexors during the test at 240°/s velocity (W) were measured. The scales for the values of the obtained parameters corresponding to the relation to the body weight and the differences between the limbs have been established according to the published standards [17]. The results for PT/BW for the extensors were classified as: insufficient (<200%), sufficient (200–249%), good (250–299%), very good (300–349%), and excellent (>350%). The results for PT/BW for the flexors were classified as: insufficient (<100%), sufficient (100–149%), good (150–199%), very good (200–249%), and excellent (>250%). The difference of the PT/BW parameter between the limbs were classified as bad (>20%), sufficient (11–20%), or good (<10%). The lowest values required for proper extensor muscle work are 3000 J and for flexors, 1800 J.

The subjective outcomes were assessed with the use of two well-established forms: the International Knee Documentation Committee knee ligament healing standard form (IKDC 2000) and the Lysholm knee scoring scale. The results of the IKDC Knee Index were classified as (according to the standards): very good (90–100 points), good (76–89 points), sufficient (50–75 points), or insufficient (<50 points). The results of the Lysholm scale were classified as (according to the standards): excellent (90–100 points), very good (80–89 points), good (70–79 points), sufficient (60–69 points), or insufficient (<60 points).

### 2.5. Statistical Analysis

All statistical analyses have been performed in Statistica v. 7.1. The Shapiro-Wilk test was used to assess the normality of the data distribution. Since the data distribution was not normal, all quantitative variables have been shown as median ± standard deviation. The significance of the differences between objective and subjective parameters was determined with the Wilcoxon signed rank test. The correlation between the outcomes was established with the Spearman’s rank correlation test. Statistical significance was set at *p* < 0.05.

### 2.6. Ethics Approval

All subjects gave their informed consent for inclusion before they participated in the study. The study was conducted in accordance with the Declaration of Helsinki, and the protocol was approved by the Bioethics Committee at the Karol Marcinkowski Poznań University of Medical Sciences (no. 830/11).

## 3. Results

### 3.1. Objective Knee Evaluation

The numbers of the “correct”, “incorrect”, and “bad” results of the postural priority did not differ significantly between the patients subjected to the 2-year follow-up and the control group (Figure 2).

A visible difference was noticed in the number of patients with “excellent” results from the postural strategy in the non-operated limb but not in the operated one. “Very good” results of PS were obtained by 25% and 19% in the operated limb, respectively, at 2-year follow-up and in the control group. At 2-year follow-up, there was a lower number of “good” results and a higher number of “sufficient” results in the operated limb when compared to the control group (11% vs. 19% and 23% vs. 15%, respectively).

The median value of the postural priority parameter oscillated around 49%, both for the operated and non-operated limbs in the 2-year follow-up (Table 3 and Figure 3). In the control group, the values of the PP parameter were lower (around 45%), but they were not statistically different in the dominant versus non-dominant limb. No statistically significant differences were found for the PP parameter in the dynamic test.

The median values of the postural strategy parameter reached around 3° (“good”) at the 2-year follow-up (Table 3). A significant difference in the value of the postural strategy parameter between operated limbs was noticed when comparing the 2-year follow-up with the control results, both in the operated and non-operated limbs (*p* < 0.05, Figure 2). Significantly lower results (PS of “sufficient”) were observed in the control group.

The median of the peak value of maximum knee extensor and flexor torque in relation to body mass at 60°/s velocity (the peak torque/body weight parameter) at 2-year follow-up was significantly lower for the operated limb than for the non-operated limb (*p* < 0.001, Table 4). However, similar differences were also observed in the control group for the dominant versus non-dominant limb (*p* < 0.001). The results for both limbs at the 2-year follow-up were sufficient.

The median value of the total work at the 2-year follow-up was lower than in the control group. The total work of the extensor muscles reached a critical value of 3000 J only in the control group. The total work of the extensors was significantly lower for the operated limb when compared to the non-operated one at the 2-year follow-up (*p* < 0.001). The difference between the extensors’ total work was also noticed for the control group—the dominant limbs were characterized by higher values than non-dominant ones (*p* < 0.05).

The results of the peak torque/body weight of extensors in the operated limb obtained at the 2-year follow-up were statistically significantly lower than for the control group (*p* < 0.05, Figure 4A). The results of the Peak Torque/Body Weight of flexor muscles in the operated limb obtained at 2-year follow-ups were significantly higher than for the control group (*p* < 0.05, Figure 4B), but lower for the non-operated limb (*p* < 0.05).

Statistically lower results of the total work of knee extensors were observed at the 2-year follow-up when compared to the control group (*p* < 0.001, Figure 5A). The results of the total work of knee extensors in both the operated and non-operated limbs obtained at the 2-year follow-up were statistically significantly lower than for the control group (*p* < 0.001). The results of the total work of the knee flexor muscles were similar (Figure 5B).

All objective results were justified by the second-look arthroscopic assessments (Figure 6).

### 3.2. Subjective Knee Evaluation

The mean IKDC score increased significantly from insufficient (42 points) at the baseline to good (78 points) at year 2 (*p* < 0.001, Table 5). The mean Lysholm score increased significantly from good (70 points) preoperatively to excellent (90 points) at the 2-year follow-up (*p* < 0.001).

To determine the influence of patient age at the time of operation, patients were divided into 3 subgroups: patients ages ≤ 32 years, 33–46 years, and >46 years. In all groups, a significant improvement was seen for both the IKDC and Lysholm, and there was no significant difference between the age groups.

To determine the body mass index (BMI) influence on the subjective assessments, participants were divided into the following groups: normal (18.5–24.9 BMI), overweight (25.0–29.9 BMI), obesity I (30.0–34.9 BMI), and obesity II (35.0–39.9 BMI). In all groups, a significant improvement was seen for both the IKDC and Lysholm, and there was no significant difference between the BMI groups.

### 3.3. Correlations between Subjective and Objective Tests

The Spearman Rho coefficient was employed to examine the correlation between the scales. Correlations between subjective and objective tests were investigated for the entire AMIC group. The results of the subjective IKDC assessments for the operated limb correlated well with the results of the isokinetic strength parameters (Table 6).

## 4. Discussion

The results of the present study indicated a significant improvement in all subjective outcome scores and almost all functional scores analyzed up to the 2-year follow-up. The positive effects of the AMIC were seen at the 2-year postoperative visit, with clinically significant improvement in functional outcomes. 30% of the AMIC patients obtained the results of all assessments of the proprioception as well as the muscle strength and endurance within the assumed standards 24 months after the procedure. These data confirm the hypothesis that the clinical benefits may be robust in AMIC-treated patients.

Our results are in accordance with the results of a recent AMIC meta-analysis based on 12 studies (11 level 4 studies and 1 level 1 study) that included a total of 375 patients [10]. Most patients were very satisfied with the result of the index procedure and would choose to undergo the same procedure again if needed.

Moreover, a recent meta-analysis of 2220 surgical procedures of focal chondral defects of the knee, including, except for AMIC, microfractures, osteochondral autograft transplantation, and autologous chondrocyte implantation (ACI), clearly showed that the AMIC procedure performed better overall at approximately three years’ follow-up [18]. The authors investigated multiple subjective outcome measures as well as data regarding hypertrophy and the rate of failures and revisions. This analysis added to the knowledge of important comparisons of AMIC versus other procedures. Some prominent examples of these comparisons include: (i) the study by Fossum et al. [19] found no significant differences between AMIC and ACI in terms of the Knee injury and Osteoarthritis Outcome Score (KOOS) and the Lysholm and Visual Analogue Scale (VAS); (ii) Volz et al. [20] reported more effective cartilage repair (clinical outcomes, ICRS Cartilage Injury Standard Evaluation Form-2000, magnetic resonance imaging with the MOCART, and BLOKS and WORMS scores) in AMIC compared to the microfractures; and (iii) Anders et al. [21] presented good, comparable clinical outcomes (modified Cincinnati and ICRS score) of AMIC and microfractures.

Thus far, no studies have examined proprioception or isokinetic assessment in patients subjected to the AMIC procedure. In this report, all AMIC patients had sufficient body control while maintaining the single-legged position with their eyes closed in the static proprioception evaluation test and good body control after 24 months. However, during the dynamic test, the posture control was disturbed according to the norms established by Riva [16]. The muscle strength and endurance of the quadriceps and flexors in the non-operated limb were stronger and more durable than in the operated limb. According to the standards established by Davies [17], the extensor strength of the operated limb was insufficient in relation to the body weight of the examined person, while the value generated by the flexors turned out to be sufficient. However, at the 2-year follow-up, the isokinetic values of both the extensors and flexors were sufficient.

A very important conclusion from this study is the observation that a significant part of the AMIC group obtained positive results in all measured tests when compared to the control group. The comparison of the isokinetic evaluation of the AMIC and the control group revealed no difference in the flexor and extensor muscle endurance. The comparison of proprioceptive results illustrates the positive effect of the balance, stabilization, and proprioception exercises performed by patients after the AMIC cartilage reconstruction following the described rehabilitation protocol. Rozzi and Kusano drew similar conclusions in their research aimed at revealing the influence of proprioceptive training on test results [22,23]. During the evaluation of proprioception in the static test, the control group showed an insufficient degree of control, i.e., too many torso deflections, as shown by the results of the Postural Strategy parameter, and this is the result of the lack of proprioception training. The AMIC group examined in the same test achieved very good PS results.

The IKDC index is a measure of knee joint function, where higher scores indicate higher activity and lower discomfort. Before the AMIC procedure, the function of the knee joint was insufficient. We found it very encouraging that the assessment of the knee joint function improved to a good level, indicating a rapid improvement in activity and only minor ailments in the operated knee joint. Indeed, Chen Chou et al. observed an increase in the IKDC scores 6 months after AMIC surgery, but their results were not as high as in the present study [24]. The IKDC results presented in this study were very similar to those described by Gobbi [25] and higher than the results obtained by AMIC patients in Hoburg’s study 12 months after the procedure [26]. The results of IKDC in the present study at 2-year follow-up are lower than those presented by the Victor and Vasso groups [27,28]. However, in all of the cited studies as well as in the present study, the results indicate a good and stable assessment of knee joint function according to the IKDC scale. The results of the IKDC 2000 scale correlated well with the objective isokinetic evaluation.

The Lysholm scores indicated a very good assessment of the knee joint function of the AMIC patients. The obtained results were higher than those presented by other authors. Hoburg showed good results 12 months after the AMIC procedure [26]. In Schagemann [29], Kaiser [13], and Gille [11,30] studies, the results of the Lysholm score revealed good knee function in the 1-year follow-up and very good at the 2-year follow-up.

## 5. Conclusions

AMIC is an effective and durable treatment, lasting up to 2 years post-surgically, for patients with cartilage defects in the knee. AMIC provides satisfactory results in terms of both subjective esteem of knee function and knee functional rehabilitation, which appear to be sustained in the majority of patients, according to our 2-year follow-up results.

## Figures and Tables

**Figure 1 jfb-13-00277-f001:**
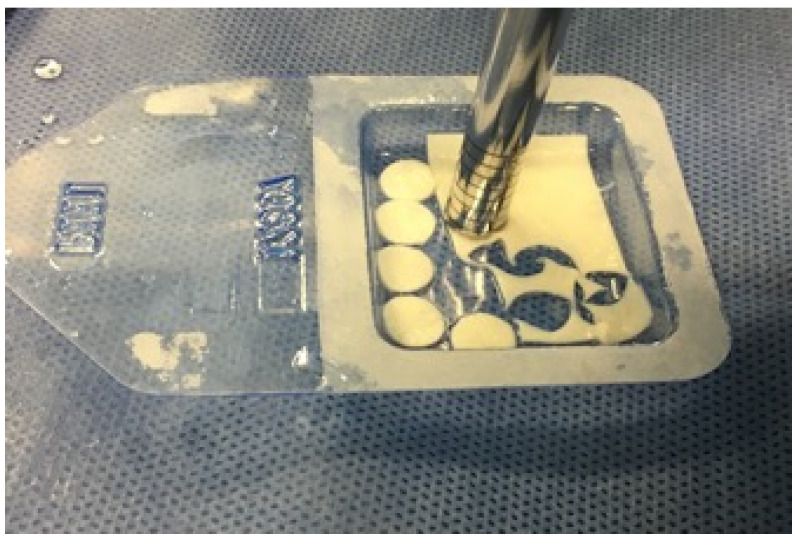
Chondro-Gide membrane circles used in the AMIC surgery.

**Figure 2 jfb-13-00277-f002:**
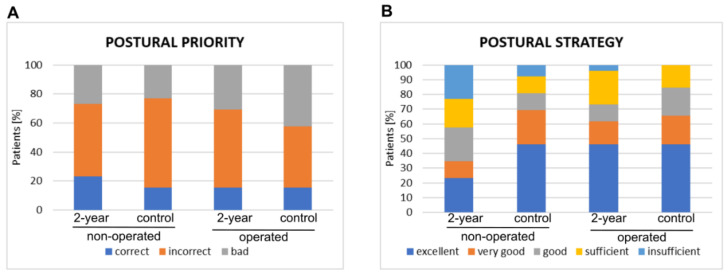
The results of the proprioceptive analysis. The number of patients with correct, incorrect, and bad results in postural priority are shown on the left (**A**). The number of patients with excellent, very good, good, sufficient, and insufficient results in postural strategy are shown on the right (**B**).

**Figure 3 jfb-13-00277-f003:**
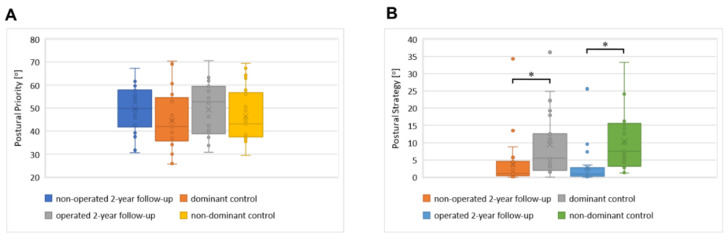
The distribution of the proprioceptive measurements. Box plot diagrams showing the distributions of the proprioceptive parameters. Central lines represent the medians, boxes indicate the range from the 25th to the 75th percentile, whiskers extend 1.5 times the above interquartile range, and outliers are represented as dots. Significance was designated as * *p* ˂ 0.05. (**A**) Postural priority results. (**B**) Postural strategy results.

**Figure 4 jfb-13-00277-f004:**
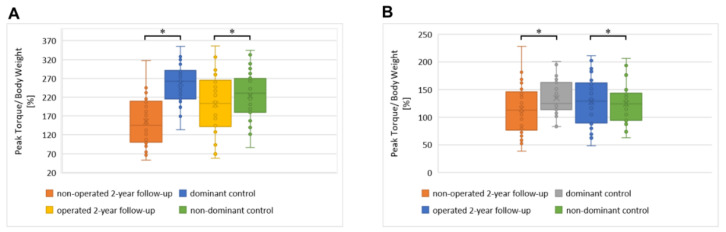
The distribution of the peak value of maximum knee extensor (**A**) and flexor (**B**) torque in relation to body mass at a 60°/s velocity. Box plot diagrams showing the distributions of the parameters. Central lines represent the medians, boxes indicate the range from the 25th to the 75th percentile, whiskers extend 1.5 times the above interquartile range, and outliers are represented as dots. Significance was designated as * with a *p* ˂ 0.05.

**Figure 5 jfb-13-00277-f005:**
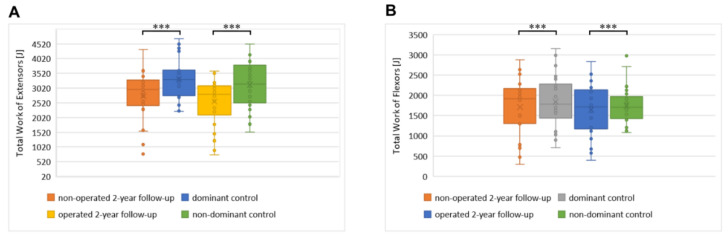
The distribution of the total work of knee extensors (**A**) and flexors (**B**) during the test at 240°/s velocity. Box plot diagrams showing the distributions of the parameters. Central lines represent the medians, boxes indicate the range from the 25th to the 75th percentile, whiskers extend 1.5 times the above interquartile range, and outliers are represented as dots. Significance was designated as *** with a *p* ˂ 0.001.

**Figure 6 jfb-13-00277-f006:**
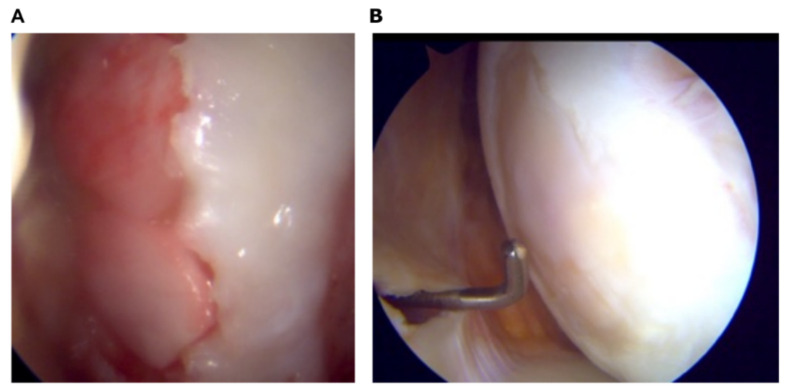
A representative second-look arthroscopic assessment of the AMIC patient before the surgery (**A**) and 2 months postoperatively (**B**).

**Table 1 jfb-13-00277-t001:** Demographic parameters of the research and a control group.

	Research Group	Control Group	*p*
Age (y.o.)	44.5 ± 11.7 (20–65)	39.1 ± 11.01 (20–65)	n.s.
Body height (cm)	174.0 ± 10.0 (155–194)	181.0 ± 6.2 (173–194)	n.s.
Body mass (kg)	82.0 ± 17.7 (59–110)	83.5 ± 13.3 (65–110)	n.s.
BMI (kg/m^2^)	26.9 ± 4.4 (20–37)	25.4 ± 3.7 (19–37)	n.s.

Mean ± standard deviation, and minimal and maximal values are presented in parentheses.

**Table 2 jfb-13-00277-t002:** The rehabilitation program for patients after the AMIC cartilage reconstruction.

Stage	Rehabilitation Program
I (until the 2nd week)	anticoagulant exercises isometric exercises range of motion-related exercises
II (3rd–8th week)	anticoagulant exercises active workout in horizontal position hip exercises (standing position on one leg) core stabilization exercises exercises with the ball: driving the ball into the wall with operated limb in horizontal position exercises in the swimming pool
III (9th–12th week)	standing exercises dynamometric platforms full-load proprioception exercises squats up to 90 degrees on an unstable ground treadmill exercises
IV (13th–24th week)	external load warming up on a stationary bike with an increasing load closed chains exercises exercises on one leg on a stable and unstable surface jumping on both legs and on one leg on a trampoline with stops
V (from 24th week)	jumping exercises aimed at increasing dynamics and power, e.g., double-leg jumps, single-leg jumps and plyometric exercises

**Table 3 jfb-13-00277-t003:** The results of the proprioceptive evaluations.

	2-Year Follow-Up			Control		
	non-operated	operated	*p*	dominant	non-dominant	*p*
**PP**	49.8 ± 9.8	49.3 ± 11.0	n.s.	44.5 ± 11.6	46.3 ± 10.6	n.s.
**PS**	3.6 ± 7.0	2.7 ± 5.3	n.s.	9.3 ± 9.8	9.6 ± 8.3	n.s.

Data are presented as mean values  ±  standard deviation. PP: postural priority, PS: postural strategy.

**Table 4 jfb-13-00277-t004:** The results of the isokinetic evaluation.

	2-Years Follow-Up			Control		
	Extensor Muscles					
	**non-operated**	**operated**	** *p* **	**dominant**	**non-dominant**	** *p* **
**PT/BW**	242 ± 80	203 ± 79	<0.001	263 ± 52	230 ± 66	<0.001
**W**	2980 ± 876	2814 ± 822	<0.001	3310 ± 702	3153 ± 795	<0.05
	**Flexor Muscles**					
	**non-operated**	**operated**	** *p* **	**dominant**	**non-dominant**	** *p* **
**PT/BW**	139 ± 46	129 ± 45	<0.05	125 ± 32	123 ± 37	<0.001
**W**	1916 ± 671	1712 ± 630	n.s.	1784 ± 649	1699 ± 454	n.s.

Data are presented as mean values  ±  standard deviation. PT/BW: peak torque/body weight [%], W: total work (J).

**Table 5 jfb-13-00277-t005:** IKDC 2000 and Lysholm scores evaluated pre- and post-operatively.

	Baseline	2-Year Follow-Up	*p*
**IKDC**	42 ± 14	78 ± 14	<0.001
**Lysholm**	70 ± 11	90 ± 7	<0.001

Data presented as mean ± standard deviation.

**Table 6 jfb-13-00277-t006:** Subjective-objective tests of correlation.

		2-Year Follow-Up	
**Subjective Test**	**Objective Test**	**r**	** *p* **
IKDC	PT/BW Ext	0.78	<0.001
IKDC	W Ext	0.59	<0.05
IKDC	PT/BW Flx	0.63	<0.01
IKDC	W Flx	0.53	<0.05
Lysholm	PT/BW Ext	0.32	n.s.
Lysholm	W Ext	0.29	n.s.
Lysholm	PT/BW Flx	0.31	n.s.
Lysholm	W Flx	0.39	n.s.

r—Spearman Rho coefficient. *p*-values are indicated.

## Data Availability

The data presented in this study are available on request from the corresponding author.

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
