# Peer review of "Autologous Matrix-Induced Chondrogenesis (AMIC) for Focal Chondral Lesions of the Knee: A 2-Year Follow-Up of Clinical, Proprioceptive, and Isokinetic Evaluation"

_jfb, 2022, doi:10.3390/jfb13040277_

Round 1

Reviewer 1 Report

The proposed study by BÄ…kowski et. al. tackles the clinical outcomes of the AMIC for the treatment of focal chondral defects of the knee, with innovation in evaluating both the subjective and objective outcomes. The study is performed on a sufficient period to draw some conclusions and the performed evaluations are in accordance with the declared aim.

L86: From a total of 61, 33 were not eligible, so a total of 48 were selected. Is it 28 maybe?

L87: “26 patients were and enrolled” - please check if correct

For the Evaluations part, are there any references to be given for the proposed classifications of the results? These values are generally known and considered or is it just a system created by the authors for this particular study?

Since the Discussion part is not too detailed, I suggest creating an independent Conclusion section.

The reference list seems rather short and outdated, please adjust it.

Author Response

Reviewer 1

The proposed study by BÄ…kowski et. al. tackles the clinical outcomes of the AMIC for the treatment of focal chondral defects of the knee, with innovation in evaluating both the subjective and objective outcomes. The study is performed on a sufficient period to draw some conclusions and the performed evaluations are in accordance with the declared aim.

L86: From a total of 61, 33 were not eligible, so a total of 48 were selected. Is it 28 maybe?

Our response: We are very sorry for this mistake, 13 patients (and not 33) were no eligible – we have changed that in the manuscript.

L87: “26 patients were and enrolled” - please check if correct

Our response: It is correct.

For the Evaluations part, are there any references to be given for the proposed classifications of the results? These values are generally known and considered or is it just a system created by the authors for this particular study?

Our response: All classifications of the results are according to the well-established, known standards and the references are cited or mentioned - this information is included in paragraph 2.4. However, because of the Reviewer’s question, we have added an explanatory sentences in mentioned paragraph.

Since the Discussion part is not too detailed, I suggest creating an independent Conclusion section.

Our response: We have created a Conclusions section, as suggested.

The reference list seems rather short and outdated, please adjust it.

Our response: We have added new, appropriate references, as suggested.

Reviewer 2 Report

Could you show the image of post operative pictures?

Author Response

Reviewer 2

Could you show the image of post operative pictures?

Our response: We have added the figure with representative pictures of a second-look arthroscopic assessment of the AMIC patient before the surgery and 2 months postoperatively, as suggested.

Reviewer 3 Report

The manuscript is well organized, documented and written. It will clearly add new data to literature. I recommend acceptance in its current form

Author Response

Reviewer 3

The manuscript is well organized, documented and written. It will clearly add new data to literature. I recommend acceptance in its current form.

Our response: Thank you for that comment.

Reviewer 4 Report

The study is focused on the functional outcomes of patients treated with Autologous Matrix-Induced Chondrogenesis (AMIC) for focal chondral lesions of the knee.

The study has a very precise focus and answers the questions posed very well. I think the study is well written and methodologically flawless. I really appreciated the description of the methods, in particular. Also, the introduction is comprehensive and the results are clear. The conclusions are consistent with the results.

I have only a few requests for revisions to improve the work, in my opinion:
1) please introduce the word "knee" in the title;
2) an intraoperative image of the procedure, particularly of the membrane used, would be helpful to readers;
3) in discussion, please include a paragraph clarifying the functional results that have emerged from studies that have investigated the use of other techniques indicated for this type of lesion ( according to size and site); especially paying special attention to comparative studies.

Thank you.

Author Response

Reviewer 4

The study is focused on the functional outcomes of patients treated with Autologous Matrix-Induced Chondrogenesis (AMIC) for focal chondral lesions of the knee.

The study has a very precise focus and answers the questions posed very well. I think the study is well written and methodologically flawless. I really appreciated the description of the methods, in particular. Also, the introduction is comprehensive and the results are clear. The conclusions are consistent with the results.

I have only a few requests for revisions to improve the work, in my opinion:

1) please introduce the word "knee" in the title;

Our response: We have added the word “knee” in the title, as suggested.

2) an intraoperative image of the procedure, particularly of the membrane used, would be helpful to readers;

Our response: All intraoperative images of the procedure are presented already in our previous publication, which precisely describes the procedure – we have added this explanatory sentence in the text. However, we have included two pictures:

Figure 1. Chondro-Gide membrane circles used in the AMIC surgery.

Figure 6. A representative of a second-look arthroscopic assessment of the AMIC patient before the surgery (A) and 2 months postoperatively (B).

3) in discussion, please include a paragraph clarifying the functional results that have emerged from studies that have investigated the use of other techniques indicated for this type of lesion (according to size and site); especially paying special attention to comparative studies.

Our response: We have included the paragraph concerning the above mentioned issues, as suggested.